# Gravitational Waves from Strange Star Core–Crust Oscillation

**Ze-Cheng Zou** [1] , **Yong-Feng Huang** [1,2,*] and **Xiao-Li Zhang** [3]

1 School of Astronomy and Space Science, Nanjing University, Nanjing 210023, China
2 Key Laboratory of Modern Astronomy and Astrophysics (Nanjing University), Ministry of Education, Nanjing 210023, China
3 School of Physics, Nanjing University, Nanjing 210023, China
* Correspondence: hyf@nju.edu.cn

**Abstract:** According to the strange quark matter hypothesis, pulsars may actually be strange stars composed of self-bound strange quark matter. The normal matter crust of a strange star, unlike that of a normal neutron star, is supported by a strong electric field. A gap is then presented between the crust and the strange quark core. Therefore, peculiar core–crust oscillation may occur in a strange star, which can produce distinctive gravitational waves. In this paper, the waveforms of such gravitational waves are derived using a rigid model. We find that the gravitational waves are extremely weak and undetectable, even for the next-generation detectors. Therefore, the seismology of a strange star is not affected by the core–crust oscillation. Observers will have to search for other effects to diagnose the existence of the crust.

**Keywords:** gravitational waves; stars: neutron; stars: oscillations

## 1. Introduction

Various oscillation modes of neutron stars are gaining astrophysical interest because they can provide information on the stellar properties [1,2]. For example, the $f$-mode oscillations of a neutron star are governed by the bulk properties, e.g., average density and radius [3], while the $g$-mode oscillations, with the restoring force being buoyancy, can reflect the interior structure [4]. As the gravitational waves (GWs) from the binary neutron star merger GW170817 have been detected [5] (which have already been used to put some constraints on the neutron star interior structure [6]), it is expected that people can directly detect the GWs from various neutron star oscillation modes and constrain neutron star properties [7].

From a microphysical perspective, the structure of neutron stars is determined by the dense matter equation of state. According to the strange quark matter hypothesis, the true ground state of strong-interaction matter may be strange quark matter consisting of nearly equal numbers of up, down, and strange quarks [8–13]. As a result, the observed pulsars may actually be strange stars [14,15]. Strange quark matter is self-bound by the strong interaction. Therefore, a bare strange star has a sharp quark surface, with the density dropping to zero in ∼fm (a typical length scale of strong interaction). On the other hand, electrons near the surface are bound by electromagnetic interaction, and thus have much more extended distribution. As a result, a strong electric field of $\sim 10^{17}\,\mathrm{V\,cm^{-1}}$ will form near the surface of a bare strange star [15].

Despite their different structures, a $1.4\,\mathrm{M_\odot}$ strange star has a very similar radius to a normal neutron star of the same mass, making these two kinds of objects hard to distinguish. What makes the problem even harder is that the strong electric field can hold a thin normal-matter crust $\sim 10^3$ fm above the surface of a strange star. The crust has a maximum mass of $\sim 10^{-5}\,\mathrm{M_\odot}$ and a maximum density well below the neutron drip density due to a detailed balance between electric and mechanical forces [16–18]. Therefore, the surface properties of a crusted strange star are also similar to those of a neutron star. Since

a strange star can accrete materials from the environment after it is born, a normal-matter crust may be naturally formed. In the following points, we will mainly focus on crusted strange stars.

Great efforts have been made to distinguish strange stars from normal neutron stars and diagnose the dense matter equation of state. It has been realized that strange stars may have a different mass-radius relation [9]. The high compactness may also enable strange stars to rotate slightly faster than neutron stars [19]. Moreover, strange stars and neutron stars are expected to have different cooling rates [20] and tidal deformability [21]. Due to the self-bound nature of strange quark matter, strange dwarfs and strange planets can even stably exist. Recently, it has been proposed that these exotic compact objects can be an interesting tool to help clarify the existence of strange quark matter. For example, a close-in planet with an orbital radius less than $\sim 5.6 \times 10^{10}$ cm or an orbital period less than $\sim 6100$ s must be a strange planet, because a normal one will be tidal disrupted at such a short distance [22,23]. The compactness criterion also stands for strange dwarfs, which distinguish themselves from white dwarfs by their smaller radii [24]. GWs from binary systems consisting of strange planets or strange dwarfs are detectable [25–27], and could carry valuable information on their tidal deformability [28].

Oscillations of strange stars can give new insights into the dense matter equation of state. The oscillation spectra, e.g., the $f$-mode spectra, are quite different between a strange star and a neutron star [29–32]. Moreover, the damping rate of the oscillation modes is faster in quark matter than in $\pi$-condensate matter, another way to distinguish between strange stars and neutron stars [33]. Since oscillating strange stars are efficient GW emitters [29,32], GWs can be a useful tool to probe their interiors.

The core–crust interface of a neutron star is somehow ambiguous, where the physical quantities are continuous. As a result, the core–crust transition point is usually manually defined by a given density [34]. In contrast, the core–crust dichotomy of a strange star is clearly established through a void gap due to the electric field. Therefore, a relative oscillation between the crust and core can happen for a strange star. In this study, we investigate this peculiar oscillation and the resulting GW emission. The oscillation model is described in Section 2. The corresponding GWs are analyzed in Section 3. Finally, we briefly summarize and discuss our results in Section 4.

## 2. Core–Crust Oscillation

We treat both the crust and core of a strange star as rigid bodies. At the equilibrium position, the centers of mass of the crust and core coincide. When a bulk perturbation is exerted on the crust or the core [35], a relative displacement ($d$) between the centers of mass of the two bodies will be induced, leading to core–crust oscillation.

Since the gap width is much smaller than the stellar radius, the gravity is nearly uniform in the gap. In contrast, the electric field drops quickly from $\sim 10^{17}$ V cm$^{-1}$ at the strange core surface to several magnitudes lower at the bottom of the crust [18]. Therefore, as the gap between the crust and core gets narrower on one side, the electric force dominates over the gravity. At the same time, on the opposite side, the gap will become wider and the gravity will dominate over the electric force. Under the effect of these net forces, the crust tends to recover its equilibrium position. As a result, a relative oscillation takes place between the crust and the core, with the net force of gravity and electric force being the restoring force. The geometry and the keyframes of the oscillation are sketched in Figure 1. Although the exact dynamics are very complex, a simple harmonic motion shall be a leading-order approximation. The relative displacement thus oscillates with an angular frequency $\omega$ as

$$d = a\cos(\omega t), \tag{1}$$

where $t$ is time. The amplitude $a$ should be smaller than the gap width, i.e., $a \lesssim 10^3$ fm.

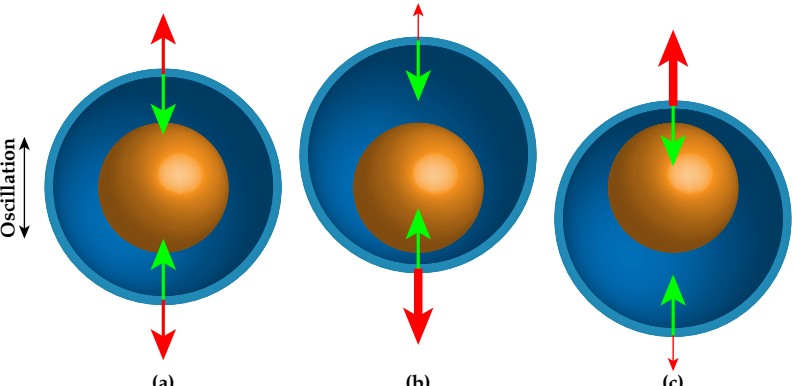

**Figure 1.** A not-to-scale sketch of the strange star core–crust oscillation. The strange quark core is shown in orange, while the normal-matter crust is shown in blue. Gravity on the crust is represented by inwards-pointing green arrows, while the electric force is represented by outwards-pointing red arrows. (**a**) An undisturbed strange star. The crust and core are at the equilibrium position, where the centers of mass of the crust and core coincide, and the gravity balances with the electric force. (**b**) When the crust deviates from the equilibrium position, its center no longer coincides with that of the core. The net force of gravity and electric force tends to reduce the displacement. (**c**) Similar to Panel (**b**), when the crust is on the other side of the equilibrium position, the net force also tends to recover the equilibrium. As a result, a relative core–crust oscillation may occur.

The frequency is a crucial parameter for the oscillation. As long as the rigid approximation is valid, the oscillation period is restricted to be longer than the perturbation propagating timescale, i.e., $2\pi/\omega > 2R_\star/c_s$, where $c_s$ is the sound speed and $R_\star$ is the stellar radius. Adopting typical values of $c_s = c/\sqrt{3}$ and $R_\star \sim 10\,\text{km}$ for strange stars [36], we have

$$\omega < 54.4\left(\frac{10\,\text{km}}{R_\star}\right)\text{kHz}, \tag{2}$$

where $c$ is the speed of light. The exact oscillation frequency depends on the detailed balance between the electric and mechanical forces. Since the electric field is induced by the electrons in the core–crust gap, the compressibility of the in-between electron layer will affect the oscillation frequency. However, during an oscillation period, the electron distribution is highly dynamic and the calculation of the compressibility will be very complicated, which is beyond the scope of this study. Regardless, Equation (2) is a strict upper bound of the frequency for any possible core–crust oscillations, so the compressibility will not markedly affect our analysis below.

## 3. Gravitational Wave Emission

GW emission occurs due to the variation of the mass quadrupole. It has two polarization states, which are usually designated $h_+$ ("plus" polarization) and $h_\times$ ("cross" polarization). For GWs propagating along the $x_3$-axis, the amplitudes of these two states read [37]

$$h_+ = \frac{1}{r}\frac{G}{c^4}\left(\ddot{I}_{11} - \ddot{I}_{22}\right), \tag{3}$$

$$h_\times = \frac{2}{r}\frac{G}{c^4}\ddot{I}_{12}, \tag{4}$$

where $G$ is the gravitational constant, $r$ is the luminosity distance of the GW source, and $I_{ij}$ is the mass second moment. In this paper, we focus on the strange star core–crust oscillation and calculate the amplitude of the resulting GWs.

To obtain the GW waveforms, one must first calculate $\ddot{I}_{ij}$ for an oscillating strange star. However, such a calculation is quite complex if one directly solves the integration $I_{ij} = \int x_i x_j \rho \, dx^3$ for the whole star, which loses its spherical symmetry in the oscillation.

Therefore, we first prove the generalized parallel axis theorem (Section 3.1) and then use it to simplify the calculation of $\ddot{I}_{ij}$ (Section 3.2).

### 3.1. Generalized Parallel Axis Theorem

In this section, we aim to find a transformation, by which one can divide $I_{ij}$ into several easy-to-calculate parts. Note that our result (Equation (7)) below is, however, universal to one's choice of division.

Considering an arbitrary mass distribution of $\rho(t, x_1, x_2, x_3)$, let us divide it into $n$ parts, i.e., $\rho(t, x_1, x_2, x_3) = \sum_{l=1}^{n} \rho^{(l)}(t, x_1, x_2, x_3)$. The total mass second moment is, therefore, a sum of the mass second moment of each part, i.e.,

$$I_{ij} = \int x_i x_j \sum_{l=1}^{n} \rho^{(l)} \, \mathrm{d}x^3 = \sum_{l=1}^{n} \int x_i x_j \, \mathrm{d}m^{(l)} = \sum_{l=1}^{n} I_{ij}^{(l)}. \tag{5}$$

The superscript $(l)$ of a variable indicates that we are calculating the physical characteristic of the $l$-th part.

For each part of the mass distribution, we first shift the origin of coordinates to its center of mass, then the new coordinate reads $x_i'^{(l)} = x_i - o_i^{(l)}$, where $o_i^{(l)}$ is the coordinate of the center of mass of this part under the old coordinate system. Here, when a variable is calculated under the new center-of-mass coordinates, a prime (') is appended; thus $o_i'^{(l)} = 0$. Therefore, the mass second moment is re-expressed as

$$\begin{aligned} I_{ij}^{(l)} &= \int \left( x_i'^{(l)} + o_i^{(l)} \right) \left( x_j'^{(l)} + o_j^{(l)} \right) \mathrm{d}m^{(l)} \\ &= \int x_i'^{(l)} x_j'^{(l)} \, \mathrm{d}m^{(l)} + \int o_i^{(l)} o_j^{(l)} \, \mathrm{d}m^{(l)} + \int x_i'^{(l)} o_j^{(l)} \, \mathrm{d}m^{(l)} + \int x_j'^{(l)} o_i^{(l)} \, \mathrm{d}m^{(l)}. \end{aligned} \tag{6}$$

The first term is the new mass second moment $I_{ij}'^{(l)}$ calculated in the new coordinates. The second term represents the mass second moment of a point mass of $M^{(l)} = \int \mathrm{d}m^{(l)}$ located at the center of mass. The last two terms vanish due to the definition of center of mass (calculated under the new coordinates) as $\int x_i'^{(l)} \, \mathrm{d}m^{(l)} = o_i'^{(l)} M^{(l)} = 0$.

Summing up all the $I_{ij}^{(l)}$, one obtains

$$I_{ij} = \sum_{l=1}^{n} I_{ij}'^{(l)} + \sum_{l=1}^{n} o_i^{(l)} o_j^{(l)} M^{(l)}. \tag{7}$$

This equation indicates that the mass second moment of any mass distribution arbitrarily divided into several parts can be decomposed into two components: (i) the sum of the mass second moments relative to the center of mass of each part; (ii) the mass second moment of several point masses, each having a mass that equals the total mass of the part.

The above result is similar to the parallel axis theorem in rigid-body dynamics [38]. Therefore, we called our Equation (7) the generalized parallel axis theorem. It is interesting to note that some specific solutions have been given for several particular cases concerning GW emission, such as two uniform rotating stars in a circular orbit [38], a point mass moving around an ellipsoidal object in a circular orbit [39], and the in-spirals of compact objects in binary common-envelope evolution [40,41]. Here, we have derived the general solution explicitly for any kind of mass (including time dependent) distribution.

### 3.2. Applied to the Core–Crust Oscillation

Considering the generalized parallel axis theorem mentioned above, we divide the strange star into two parts ($n = 2$), i.e., the strange core and the crust. According to Equation (7), we have

$$I_{ij} = I_{ij}'^{\text{core}} + I_{ij}'^{\text{crust}} + o_i^{\text{core}} o_j^{\text{core}} M^{\text{core}} + o_i^{\text{crust}} o_j^{\text{crust}} M^{\text{crust}}. \tag{8}$$

Equivalently speaking, we have divided the mass second moment of a strange star into three parts , i.e., (a) the moment of the strange core relative to its center of mass $I_{ij}^{\prime\text{core}}$, (b) the moment of the crust relative to its center of mass $I_{ij}^{\prime\text{crust}}$, and (c) the moment of a two-point-mass system, with one point mass $M^{\text{core}}$ representing the strange core and the other $M^{\text{crust}}$ corresponding to the crust, respectively. According to Equations (3) and (4), only the second time derivatives of the mass second moment contribute to the mass quadrupole variation and GW emission. Under the rigid approximation, the second time derivatives of terms (a) and (b) vanish as $\ddot{I}_{ij}^{\prime\text{core}} = \ddot{I}_{ij}^{\prime\text{crust}} = 0$; thus, the GWs produced by strange star core–crust oscillation are identical to the case of two oscillating point masses $M^{\text{core}}$ and $M^{\text{crust}}$ located at the centers of mass of the core and the crust, respectively, separated by a distance $d$ (Equation (1)), i.e.,

$$\ddot{I}_{ij} = \frac{\mathrm{d}^2}{\mathrm{d}t^2}\left(o_i^{\text{core}}o_j^{\text{core}}M^{\text{core}} + o_i^{\text{crust}}o_j^{\text{crust}}M^{\text{crust}}\right). \tag{9}$$

The overall GW waveforms from two oscillating point masses are well studied, and read [37]

$$h_+ = \frac{2G\mu a^2\omega^2}{rc^4}\sin^2\theta\cos(2\omega t_{\text{ret}}), \tag{10}$$

$$h_\times = 0, \tag{11}$$

where $\mu = M^{\text{core}}M^{\text{crust}}/(M^{\text{core}} + M^{\text{crust}})$ is the reduced mass of the core–crust system, $t_{\text{ret}}$ is the retarded time, and $\theta$ is the viewing angle between the line of sight and the oscillating direction. The resulting GWs have a frequency of $f_{\text{GW}} = \omega/\pi$.

Taking typical parameters, the GW amplitude is

$$h \approx 10^{-51}\left(\frac{1\,\text{kpc}}{r}\right)\left(\frac{\mu}{10^{-5}\,\text{M}_\odot}\right)\left(\frac{a}{10^3\,\text{fm}}\right)^2\left(\frac{f_{\text{GW}}}{3\,\text{kHz}}\right)^2. \tag{12}$$

The assumed GW frequency corresponds to an oscillation angular frequency of $\omega \approx 9\,\text{kHz}$, which fulfills the necessary condition of the rigid approximation (see Equation (2)). The amplitude of $\sim 10^{-51}$ is very low; therefore, such GWs are very unlikely to be detected.

Interestingly, persistent GW emissions have previously been studied by several other groups [29–32]. Note that these earlier studies concentrated on different oscillation modes, such as the $f$-mode or $r$-mode, which differs markedly from our oscillation mode. Generally, GW emissions of the $f$-mode or $r$-mode are much stronger than the power from the core–crust oscillation considered here. In our case, the small value of $h$ in Equation (12) is mainly due to the small oscillation amplitude $a$, which is restricted by the gap width. Energy of $\sim 10^{-9}\,\text{M}_\odot c^2$ is required in the $f$-mode oscillations to get a detectable GW of $h \sim 4 \times 10^{-22}$ at a distance of 1 kpc [7]. In contrast, the total mechanical energy of the crust is only $E = \mu a^2\omega^2/2 \approx 5 \times 10^{-39}\,\text{M}_\odot c^2$ in the case of core–crust oscillation, which is definitely too small. Therefore, a much lower GW amplitude is expected in our case. However, note that the core–crust oscillation may coexist with the $f$-mode and $r$-mode oscillations, which may make the problem even more complicated, and it may need to be further studied in the future.

To quantitatively assess the detectability of the GWs by a particular GW detector, it is useful to introduce the strain spectral amplitude $h_f$ of GWs [36,42,43]. Under the optimistic condition that the GWs keep monochromic and constant in the amplitude (see below for the GW damping timescale), $h_f$ can be calculated as

$$h_f \simeq h\sqrt{T} \approx 10^{-47}\left(\frac{1\,\text{kpc}}{r}\right)\left(\frac{\mu}{10^{-5}\,\text{M}_\odot}\right)\left(\frac{a}{10^3\,\text{fm}}\right)^2\left(\frac{f_{\text{GW}}}{3\,\text{kHz}}\right)^2\left(\frac{T}{5\,\text{yr}}\right)^{1/2}\,\text{Hz}^{-1/2}, \tag{13}$$

where a typical observation time of $T = 5\,\text{yr}$ is assumed for the next-generation Einstein Telescope[1]. Compared to the $\sim 10^{-24}\,\text{Hz}^{-1/2}$ sensitivity of the Einstein Telescope at kHz frequency, the GWs from a strange star core–crust oscillation are clearly undetectable.

Notwithstanding the undetectability, whether the GW emission can affect the dynamics of the oscillation itself should be considered. The gravitational radiation has a luminosity of [37]

$$P = \frac{16}{15}\frac{G\mu^2}{c^5}a^4\omega^6. \tag{14}$$

With total mechanical energy of $E = \mu a^2\omega^2/2$, the GW damping timescale is estimated as

$$\tau \sim \frac{E}{P} \approx 3 \times 10^{27}\left(\frac{10^{-5}\,\text{M}_\odot}{\mu}\right)\left(\frac{10^3\,\text{fm}}{a}\right)^2\left(\frac{10\,\text{kHz}}{\omega}\right)^4\,\text{yr}. \tag{15}$$

We see that the timescale is much longer than the Hubble time, indicating that the GW damping is very inefficient.

## 4. Conclusions and Discussion

In this paper, the relative oscillation between the crust and core of a strange star is studied, and the resulting GWs are investigated in detail. Using the generalized parallel axis theorem, we argue that the GW waveforms are identical to that of the oscillation of two point masses. The resultant GWs are found to be too weak even for next-generation GW detectors such as the Einstein Telescope. As a result, even if the strange star undergoes some kinds of stellar oscillations, the relative oscillation between its crust and core has a negligible contribution to the overall GWs from the total seismology of the strange star. It is also found that such a core–crust oscillation cannot be damped efficiently by the GW emission. Since the electric field is strong in the core–crust gap, the electromagnetic damping may be more important and may lead to some other effects, which are beyond the scope of the current study.

In our calculations, for simplicity, we assume the electron layer in the gap is compressible and neglect the electron degenerate pressure. For a more accurate study, one should consider the degenerate pressure as well as its gradient. The response of the electron layer to the external perturbation should also be examined. However, since our Equation (2) is a strict upper limit for the oscillation frequency, our results on the GW emissions will not be significantly affected.

What triggers the core–crust oscillation remains in question. One possible source is the fractional collapse of the crust through accretion from a stellar companion [44]. The accreted materials will flow along the magnetic field lines and accumulate near the polar cap. When the crust there becomes too heavy to be supported by the electric field in the gap, a fractional collapse occurs [45]. The falling materials will hit and merge with the core, transferring its momenta and causing a displacement. Another possibility is the collision of an asteroid or comet with the strange star [44,46]. In both scenarios, the trigger of the core–crust oscillation may be accompanied by various phenomena such as fast radio bursts [45,47,48] or pulsar (anti-)glitches [46]. Tidal interactions in a binary, although believed to be able to trigger other resonant oscillations [34], may be inefficient to produce a relative displacement between the crust and core.

GWs at kilohertz are a promising tool when it comes to probing the compact star structure and the dense matter equation of state [49]. Inspirals of compact binaries can give novel insights into the study of exotic strange quark matter objects [25–28,36]. GW signals from compact star seismology can also place constraints on the strange star properties. The $f$-mode oscillations, first introduced by Lord Kelvin [50] to consider the simple case of uniform density distribution, may be more hopeful for observational tests. We hope that future GW observations on compact star oscillations can greatly improve our limited knowledge of the extreme physics deep inside compact stars.

**Author Contributions:** Conceptualization, Y.-F.H.; methodology, Y.-F.H., Z.-C.Z. and X.-L.Z.; software, Z.-C.Z.; validation, Y.-F.H., Z.-C.Z. and X.-L.Z.; formal analysis, Z.-C.Z.; investigation, Z.-C.Z.; resources, Y.-F.H. and Z.-C.Z.; data curation, Y.-F.H., Z.-C.Z. and X.-L.Z.; writing—original draft preparation, Z.-C.Z.; writing—review and editing, Y.-F.H. and X.-L.Z.; visualization, Z.-C.Z.; supervision, Y.-F.H.; project administration, Y.-F.H.; funding acquisition, Y.-F.H. All authors have read and agreed to the published version of the manuscript.

**Funding:** This work is supported by the National Natural Science Foundation of China (Grant Nos. 12233002, 12041306, 11873030, 12147103, and U1938201), the National Key R&D Program of China (2021YFA0718500), the National SKA Program of China No. 2020SKA0120300, and science research grants from the China Manned Space Project with No. CMS-CSST-2021-B11.

**Institutional Review Board Statement:** Not applicable.

**Informed Consent Statement:** Not applicable.

**Data Availability Statement:** This study did not report any new data.

**Acknowledgments:** We thank the referees for valuable suggestions that led to an overall improvement in this study.

**Conflicts of Interest:** The authors declare no conflict of interest.

## Note

1　　https://apps.et-gw.eu/tds/ql/?c=15418 (accessed on 28 May 2022).

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
