# Peer review of "Gravitational Waves from Strange Star Core–Crust Oscillation"

_universe, doi:10.3390/universe8090442_

Round 1

Reviewer 1 Report

I am surprised to read in Equation 10 this small value for the GW amplitude of strange stars. What the authors claim should seem at odds with the article by Anderson et al 2001. I wonder if the authors are not missing some physical effect in their model.  I would invite the authors to verify their model and explain the differences with Anderson's article.

Author Response

> I am surprised to read in Equation 10 this small value for the GW
> amplitude of strange stars. What the authors claim should seem at
> odds with the article by Anderson et al 2001. I wonder if the authors
> are not missing some physical effect in their model. I would invite
> the authors to verify their model and explain the differences with
> Anderson's article.

Re: We thank the referee for reminding us of the results obtained by
Andersson et al. We have re-examined our Equation (now labelled Eq. 12)
and guarantee that there are no mistakes in our calculations. The 
differences between our results and those of Andersson et al. are 
explained as follows. Andersson et al. considered a different oscillation
mode (r-mode) as compared with ours. We also noticed that some other 
groups have additionally considered other modes such as the f-mode. 
So, the GW emissions of our core-crust oscillation and other modes 
could be very different. There is no conflict between our results 
and other groups'. Basically, the weakness of the GW emission in our 
case is due to the much less mechanical energy stored in the 
oscillating crust. 

We have added a paragraph following Eq. 12 to explain the differences
between our work and earlier studies. Please see Lines 165 - 177 on
Page 5 for our explanations.

Reviewer 2 Report

This paper studies the gravitational wave feature from the strange star core-crust oscillation by a generalized parallel axis theorem, and finds that the gravitational waves are extremely weak and undetectable even for the next generation detectors. Although the associated gravitational wave  is undetectable, this study is of some interest for understanding the dynamics of strange stars. As a result, this study deserves publication in Universe. 

Author Response

> This paper studies the gravitational wave feature from the strange
> star core-crust oscillation by a generalized parallel axis theorem,
> and finds that the gravitational waves are extremely weak and
> undetectable even for the next generation detectors. Although the
> associated gravitational wave is undetectable, this study is of some
> interest for understanding the dynamics of strange stars. As a
> result, this study deserves publication in Universe. 

Re: We thank the referee for the very encouraging comments.

Reviewer 3 Report

The authors calculated the gravitational wave radiation of a strange star due to the separated oscillation of the stellar core and crust, where the core and crust are separated by a gap of   a width of a few thousands of femtometers. The idea is novelty and worth to be published in Universe. I have two major suggestions as follows.

(1) The gap between the stellar core and crust is full filled with electrons. The density of the electrons is determined by the mechanical equilibrium of the Coulomb attractive force and the degenerate pressure. Therefore, the stellar oscillation would also interact with this electron layer, rather than simply taking place in vacuum. So, the primary assumption of this paper is that the electron layer is an easily-compressed fluid, but the crust and core can be regarded as rigid body. In principle, the reasonability of this assumption needs to be judged carefully, in view of that the core is a fluid too although it is self-bound. Could the authors give some estimations or discussions?

(2) A crucial parameter for an oscillation is its frequency, which can directly reflect the dynamics determining the oscillation. So, is it possible to give a rough analysis on the potential frequency? Or, just discuss which factors can influence the oscillation frequency.

Author Response

> The authors calculated the gravitational wave radiation of a strange
> star due to the separated oscillation of the stellar core and crust,
> where the core and crust are separated by a gap of a width of a few
> thousands of femtometers. The idea is novelty and worth to be
> published in Universe. I have two major suggestions as follows.

Re: We thank the referee for the very helpful comments. We have revised
our manuscript accordingly. All the revisions are marked in boldface in
the new version. Please see our detailed replies to each of the
comments below.

> (1) The gap between the stellar core and crust is full filled with
> electrons. The density of the electrons is determined by the
> mechanical equilibrium of the Coulomb attractive force and the
> degenerate pressure. Therefore, the stellar oscillation would also
> interact with this electron layer, rather than simply taking place in
> vacuum. So, the primary assumption of this paper is that the electron
> layer is an easily-compressed fluid, but the crust and core can be
> regarded as rigid body. In principle, the reasonability of this
> assumption needs to be judged carefully, in view of that the core is
> a fluid too although it is self-bound. Could the authors give some
> estimations or discussions?

Re: We thank the referee for reminding us of the role of electron
degenerate pressure. In our calculations, for simplicity, we have 
neglected the electron degenerate pressure. To consider the degenerate 
pressure, we would also need to consider its gradient and the response 
of the electron layer to the external perturbations. These factors will 
be too complicated to be included in this study. At the same time, since 
our Equation 2 is a strict upper bound for the oscillation frequency, 
a more accurate calculation of the frequency will not significantly 
affect our final results on the GW emissions. Our main conclusions thus 
would not be markedly affected by these factors, as we focus on a
small-perturbation solution around the equilibrium position.

We have added a paragraph in Sect. 4 to discuss this issue. Please see
Lines 203 - 208 on Page 6 for our discussions.

> (2) A crucial parameter for an oscillation is its frequency, which
> can directly reflect the dynamics determining the oscillation. So, is
> it possible to give a rough analysis on the potential frequency? Or,
> just discuss which factors can influence the oscillation frequency.

Re: We completely agree with the referee that the frequency is a crucial
parameter for any oscillations. In our case, the exact oscillation
frequency depends on the detailed balance between the electric and
mechanical forces. However, during the oscillation, the electron
distribution in the gap is highly dynamic, and much complexity will be
introduced if considering other effects (as explained in our reply to the
previous comment). As a result, it is hard to determine the oscillation
frequency analytically. Anyway, we notice that Eq. 2 could serve as a 
strict upper bound for the oscillation frequency. So, our main conclusions
would not be affected by the frequency uncertainty.

In the revised version, we have stressed the importance of the
oscillation frequency (Line 86) as reminded by the referee. We also 
expand the paragraph around Eq. 2 to discuss the determination of 
the oscillation frequency. Please see Lines 91 - 98 on Page 3 for 
our revisions concerning this issue.

Reviewer 4 Report

The article with title "Gravitational Waves from Strange Star Core-Crust Oscillation" investigated the gravitational waves from the relative oscillation between the crust and core of a strange star. They found the GWs is not important in oscillaton damping and not detactable even for the next generation GW detectors. The topics is common interested and the results are reasonable. I recommend it to be accepted by the Universe.

Following are some comments the author might want to address:

Fig 1., the arrow sizes of the gravity, are the same for the 3 different cases. It is not obvious just from the skematic figure. The authors may explain it.

In Section 3.1, it is worth to inllustrate the meaning of the superscript (l), and how it affects the overall GW signals. Further, why a prime coordinate is needed.

Specifically, how it derives to equations 8 and 9 from equation 7? (e.g, n=?, I'_ij=?, o'_i^(l)=?, M^(l)=?)

On the other hand, as shown in section 3.2, if equation 7 is derived from equations 3 and 4, what is the motiviation for showing the entire section 3.1?

line 110, curst should be crust instead.

Author Response

> The article with title "Gravitational Waves from Strange Star
> Core-Crust Oscillation" investigated the gravitational waves from the
> relative oscillation between the crust and core of a strange star.
> They found the GWs is not important in oscillaton damping and not
> detactable even for the next generation GW detectors. The topics is
> common interested and the results are reasonable. I recommend it to
> be accepted by the Universe.

Re: We thank the referee for the very helpful comments. We have revised
our manuscript accordingly. All the revisions are marked in boldface in
the new version. Please see our detailed replies to each of the
comments below.

> Following are some comments the author might want to address: 
>
> Fig 1., the arrow sizes of the gravity, are the same for the 3
> different cases. It is not obvious just from the skematic figure. The
> authors may explain it.

Re: We thank the referee for reminding us about this point. The gravity
is uniform in the gap because the gap width is much smaller than the
stellar radius, while the electric field is not uniform. We
have added a few sentences to explain this. Please see Lines 73 - 76 on
Page 2 for our revisions.

> In Section 3.1, it is worth to inllustrate the meaning of the
> superscript (l), and how it affects the overall GW signals. Further,
> why a prime coordinate is needed.

Re: We thank the referee for reminding us that our symbols may lead to
confusion. The superscript (l) means that we are calculating the
physical parameters of the l-th part of the mass distribution. As a
summing index, it means that we are summing among all the parts
and the (l) itself does not affect GWs. Please see Lines 118 - 119 on
Page 4 for our revisions.

The prime symbol (') indicates that a variable is calculated in the
center-of-mass coordinates, which is used to help prove the generalized
parallel axis theorem. Please see Lines 122 - 124 on Page 4 for our
revisions.

> Specifically, how it derives to equations 8 and 9 from equation 7? (e.
> g, n=?, I'_ij=?, o'_i^(l)=?, M^(l)=?)

Re: We thank the referee for pointing out our lack of detailed
derivation. In the revised version, we have revised the first paragraph
of Sect. 3.2 to show the details of the derivation. Especially, we have
added two new equations 8 and 9 between Eqs. 7 and 10 & 11 (old Eqs.
8 & 9). Please see Lines 142 - 158 on Page 5 for our revisions.

> On the other hand, as shown in section 3.2, if equation 7 is derived
> from equations 3 and 4, what is the motiviation for showing the
> entire section 3.1?

Re: We thank the referee for pointing out our lack of information.  
Sect. 3.1 is used to prove Eq. 7 from the definition of Iij.
Moreover, Eq. 7 is proved with an aim to simplify the calculation of
the second mass moment. 

Please see Lines 106 - 114 on Page 4 for our revisions concerning this issue.

> line 110, curst should be crust instead.

Re: We thank the referee for pointing out the typo. In the revised
version, we have searched our text and fixed all the typos alike.